# NO_2_, BC and PM Exposure of Participants in the Polluscope Autumn 2019 Campaign in the Paris Region

**DOI:** 10.3390/toxics11030206

**Published:** 2023-02-23

**Authors:** Laura Bouillon, Valérie Gros, Mohammad Abboud, Hafsa El Hafyani, Karine Zeitouni, Stéphanie Alage, Baptiste Languille, Nicolas Bonnaire, Jean-Marc Naude, Salim Srairi, Arthur Campos Y Sansano, Anne Kauffmann

**Affiliations:** 1Laboratoire des Sciences du Climat et de l’Environnement (LSCE-IPSL), UMR CNRS-CEA-UVSQ, 91191 Gif-Sur-Yvette, France; 2Laboratoire DAVID, Université Saint-Quentin-en-Yvelines, 78035 Versailles, France; 3Department of Chemistry, University of Oslo, 0315 Oslo, Norway; 4Cerema, Île-De-France, Département Mobilité, 78190 Trappes-en-Yvelines, France; 5Airparif, 75004 Paris, France

**Keywords:** personal exposure, air quality, Île-De-France, portable sensors, urban pollution

## Abstract

The Polluscope project aims to better understand the personal exposure to air pollutants in the Paris region. This article is based on one campaign from the project, which was conducted in the autumn of 2019 and involved 63 participants equipped with portable sensors (i.e., NO_2_, BC and PM) for one week. After a phase of data curation, analyses were performed on the results from all participants, as well as on individual participants’ data for case studies. A machine learning algorithm was used to allocate the data to different environments (e.g., transportation, indoor, home, office, and outdoor). The results of the campaign showed that the participants’ exposure to air pollutants depended very much on their lifestyle and the sources of pollution that may be present in the vicinity. Individuals’ use of transportation was found to be associated with higher levels of pollutants, even when the time spent on transport was relatively short. In contrast, homes and offices were environments with the lowest concentrations of pollutants. However, some activities performed in indoor air (e.g., cooking) also showed a high levels of pollution over a relatively short period.

## 1. Introduction

Air pollution has become a significant public health concern today, responsible for approximately 50,000 deaths per year in France [1,2]. Exposure to air pollutants can be chronic or acute and lead to serious health problems, such as rhinitis and asthma [3,4,5]. Such adverse effects can occur at lower concentrations than previously determined, demonstrated by the new thresholds recommended by the World Health Organization (WHO) ambient air quality guidelines [5,6]: 10 µg·m^−3^ per year and 25 µg·m^−3^ over 24 h for nitrogen dioxide (NO_2_), 5 µg·m^−3^ per year and 15 µg·m^−3^ over 24 h for particulate matter ≤2.5 µm (PM_2.5_) and 15 µg·m^−3^ per year and 45 µg·m^−3^ over 24 h for particulate matter ≤10 µm (PM_10_) [6].

Monitoring the concentrations of the various regulated or tracked pollutants in outdoor air is generally carried out by air quality networks through ambient measurements at fixed stations, which allows concentrations to be measured with high accuracy [6,7]. Although these measurements are used for personal exposure assessments (PEs), they are not able to correctly represent the indoor pollutant concentration variability that populations are exposed to [8]. With the development of portable sensors, it has become possible to measure the concentrations of pollutants to which a person is exposed to throughout the day, depending on the environment in which they are located. Using such devices could provide a more accurate representation of their PE [9].

PE has no clear definition or standard procedure for quantification [10]. Different studies have used various definitions and techniques. Many studies have been conducted with portable sensors, and others had compared different types of sensors and their agreement with reference instruments [9]. PE has also been studied worldwide using portable sensors and fixed stations to understand the differences in exposures related to different activities and environmental conditions [11]. In addition, it has been investigated in specific environments such as the street and various modes of transportation [12,13]. The pollutants measured in the studies vary, with several focusing on particulate pollution, particularly by measuring black carbon (BC) [10,14,15,16]. Other studies have focused on assessing PM ranging from ultra-fine particles to PM_10_ [10,17,18,19,20]. One of these included 50 participants, allowing a better representation [18]. The diverse gas-phase pollutants studied, such as NO_2_ and volatile organic compounds (VOCs), presented a more accurate picture of everyday exposures [10,12,20]. These different studies have led to a better understanding of PE in various environments. However, few studies have measured multiple pollutants simultaneously, and their measurement times were generally relatively short [10].

The ANR-Polluscope project (2016–2022) aims to provide complementary information by estimating the PE of a selection of persons in Île-de-France (IDF), the most populated area in France with 12 million inhabitants [8]. The air quality network Airparif is in charge of monitoring the concentrations of regulated air pollutants in outdoor ambient air in this region. This project relied on volunteering: participants were equipped with portable sensors for one week in the autumn of 2019 [8]. These sensors measured NO_2_, BC, and PM. These air pollutants were chosen because they are subject to regulation due to their health impacts [8]. They were also chosen according to the reliability of the tested sensors.

NO_2_ is an oxidizing gas with a pungent smell that can be present in indoor and outdoor air [3,21]. This gas is generally produced indoors by high-temperature combustion activities such as gas-powered cookers or heaters and the penetration of outdoor air [22]. In IDF, NO_2_ pollution is mainly produced from outdoor vehicles [23]. PM can be present in both indoor and outdoor air in solid and/or liquid form [22,24]. Most indoor PM comes from tobacco smoke, biomass burning for heating, cooking, or from the outdoor environment when windows are open [22]. There are multiple sources of outdoor PM, but the tertiary sector represents the most significant contribution to direct emissions, followed by road traffic, construction sites, and agriculture [25,26]. A secondary, yet significant, part of the particles is derived from the oxidation of gaseous precursors or primary particles [26]. BC is a particulate pollutant that is produced by the incomplete combustion of biomass, fossil fuel combustion, or cigarette smoke [23,25]. The predominant sources of outdoor BC are cars, while indoor BC sources include heating appliances, fireplaces, lit candles, and food charring during cooking [27,28]. This pollutant is not regulated, but the French National Agency recommended its monitoring for health safety (ANSES) because it can cause respiratory problems [21,29,30].

The research conducted by Languille et al. in 2018–2019 provided valuable initial results [10]. The study was the first conducted in IDF for the simultaneously measure of PM, BC, and NO_2_ with sensors worn by volunteers. One of the key results was the strong influence of a handful of environments and activities, even though the time spent there was short. The road traffic proximity accounted for up to 34% of BC and 26% for NO_2_, while cooking and tobacco smoke accounted for up to 44% of the PE to PM (percentages mentioned here are the contributions to PE).

Although the conclusions of this study were fruitful, it nevertheless presented limitations. First, the campaign was one week and the number of participants (37 in total) was quite limited, especially for the campaign conducted in the winter of 2019. These campaigns also highlighted the fact that participants did not systematically fill in their schedules at each change of environment which made data analysis complicated. The assignment of activities was performed only according to what the volunteers indicated in their schedule logbook. In addition, most participants in the campaign were volunteers from partner institutions, i.e., with a scientific background, who agreed to test the protocol during the campaign.

Based on the lessons learned from previous work, we designed an ambitious protocol for new campaigns in order to go beyond the limitations mentioned above [10]. More precisely, this research was based on data collected with the collaboration of 63 volunteer citizens over five weeks. These citizens were recruited by responding to a call for participatory science. The issue of incomplete schedules was overcome using a machine-learning tool that we developed. Additional experiments were conducted to better characterize specific environments. The primary goal of this study was to refine the quantification and progress in understanding the PE to citizens in ÎDF.

## 2. Materials and Methods

### 2.1. Presentation of the Campaign

The Polluscope campaign took place in the autumn of 2019 in IDF. It lasted five weeks, from mid-October to mid-December, and involved 63 volunteers. The volunteers wore two or three sensors in a backpack, depending on the availability of sensors. In addition, they had a tablet at their disposal to record their different activities. The participants wore the sensors for one week. Between each week of measurements, there was a week break so that the sensors could be checked, and the data retrieved.

The AE51 (AethLabs, San Francisco, CA, USA) was used to measure BC; its measurement principle is based on infrared absorption on a filter [31]. The deposition on the filter is measured by a light-emitting diode (LED) at 880 m and a photo-diode detector. The absorbance of the deposition is compared to a blank portion of the filter, which serves as a reference. Both measurements are made with the same time step. During the campaign, only six units of AE51 were used because of their relatively high cost. The AE51 was placed in the backpack; the air was sucked through a short sampling line with a flow rate of 150 mL/min.

The Cairsens sensors (Envea Cairpol Microsensors, Poissy, France) were used to measure NO_2_ [32]. The measurement principle of this sensor is based on electrochemistry [32]. It has a miniaturized measuring cell composed of three electrodes: the anode, the cathode, and the reference electrode [32]. The electrical signal generated between the anode and the cathode is proportional to the concentration. Fifteen sensors were used during the campaign; the sensor had a dynamic air-sampling system, and it was hung outside the backpack.

The Canarin II was used to measure PM_1_, PM_2.5_, and PM_10_; its measurement principle is based on laser light scattering [33]. This sensor is the result of a joint project between the Laboratoire d’Informatique de Paris 6 located at the Université Pierre et Marie Curie in Paris, the Asian Institute of Technology in Bangkok, and the Macao Polytechnic Institute in Macao, China [33]. It uses the principle of laser scattering to irradiate the suspended particles, the scattered light is collected at an angle, and the variation curve with time is obtained. The equivalent particle diameter and the number of particles per unit volume according to the particle size can then be calculated. This sensor is equipped with a GPS and a Wi-Fi interface to send the data. Fifteen sensors were used during the campaign, and like the Cairsens, it was attached to the outside of the backpack. A more detailed description of the sensors is given by Languille et al. (2020) [34]. The time resolution was set to one minute for all sensors, which provided 3,504,418 measurement points.

### 2.2. Sensor Qualification Tests

The first phase (2017–2018) of the Polluscope project was devoted to selecting and qualifying sensors. All these tests are described by Languille et al. (2020), and therefore only a very brief summary is given here [34]. The first test performed was the fixed measurement test. The various sensors were placed at the SIRTA site for qualification. The SIRTA site is part of the ACTRIS (Aerosols, Clouds, and Trace Gases Research Infrastructure) European research infrastructure. It is a peri-urban monitoring station located about 20 km south-west of Paris. The sensors were fixed at the entrance of the SIRTA site at a height of 2.50 m in the open air under a shelter so that they were not exposed to rain. The sensors were permanently connected so that there would be no data loss due to battery failure. The test lasted one week to have a sufficiently large dataset and a good representation of the sensors’ capabilities [34]. The sensors were then tested in a controlled atmosphere and mobility conditions. All these tests are described in Languille et al. (2020).

After the selection tests (2017) and the purchase of all sensors (2018), several qualification tests were performed. Here, we show the results from the 2019 qualification (the closest to our measurement campaign period), whereas the qualification results presented by Languille et al. (2022) were performed in 2018 [10]. The Canarin II and the AE51 sensors were qualified at the SIRTA–ACTRIS site with the same protocol as the fixed measurement test performed for the selection test. The Cairsens was qualified by Airparif at a measurement station classified as a “traffic proximity” station because the NO_2_ concentrations at the SIRTA site were too low and often below the detection limit of the Cairsens sensors (20 ppb). The concentrations measured by the sensors were then compared to the measurements of reference instruments. The Cairsens was compared to the 42i analyzer (Thermo Fisher Scientific, Waltham, MA, USA); for the Canarin the PM_1_ was compared to a TEOM PM_1_ 1400 coupled to an FDMS 850 module (Thermo Fisher Scientific, Waltham, MA, USA), the PM_2.5_ to a TEOM PM_2.5_ 1400 coupled to an FDMS 850 module (Thermo Fisher Scientific, Waltham, MA, USA) and the PM_10_ to a TEOM PM_10_ 1400 coupled to an FDMS 850 module (Thermo Fisher Scientific, Waltham, MA, USA). In addition, the Canarin data were compared to the data from FIDAS 200 (PALAS, Karlsruhe, Germany), which measures all PM. The AE51 sensors were compared to the Aethalometer Model AE33-7 (MAGEE Scientific, Berkeley, CA, USA). All reference instruments had a time step of 1 min, except for the TEOM PM_1_ 1400 coupled to an FDMS 850 module and the TEOM PM_2.5_ 1400 coupled to an FDMS 850 module which had a time step of 15 min, while the TEOM PM_10_ 1400 coupled to an FDMS 850 module had a time step of 5 min; therefore, the data from the sensors had to be averaged for comparison. Most of the reference instruments are described in more detail by Petit et al. (2015) [35].

After these various tests, the evaluation algorithm developed by Fishbain et al. (2017) was used to derive a performance index (IPI) for each sensor. The closer the index is to 1, the more reliable the sensor is (Table 1). These indices are calculated according to seven different metrics: root-means-square error (RMSE), different correlation coefficients (Pearson, Kendall, Spearman), the ratio of recorded over missing data, match score, and low-frequency energy (LFE). The IPI index was then calculated for each sensor. An average value for each type of sensor for the 2019 qualification campaign (Appendix A) is given. The results of the Pearson correlation coefficients (Table 1) showed that for Canarin the uncertainty was about 10% for PM_1_ and PM_2.5_, and 20% for PM_10_ (Appendix A). For AE51, the uncertainty was also about 20% (Appendix A), whereas for the Cairsens the uncertainty was more significant, around 35% (Appendix A). Although relatively low for portable sensors, these uncertainties were estimated for fixed measurements in outdoor air. They may be higher when measurements are performed in mobility and in environments such as railway stations (see above). Another source of error could come from the influence of humidity on PM measurements, as no correction was made on PM, even at high humidity. Nevertheless, it is noteworthy that the median and P95 values of relative humidity were similar for the campaign and qualification phase (medians of 37.3% and 34.5%; and P95 of 53.9% and 48.2%, respectively). 

The Canarin II has a time resolution of about 1 min and works on the principle of an optical counter. This can lead to artifacts in mass concentration measurements. The conversion of numbers to mass involves using a density factor which may be more or less accurate depending on the chemical nature of the aerosols. In particular, PM composition is known to be higher loaded with metals in underground railway stations leading to a higher density of particles [36]. For PM_2.5_, a density between 2.2 and 3.1 has been estimated in underground stations, whereas the density is usually between 1 and 2.3 in ambient air [36,37]. Therefore, a correction factor of ~2 should be applied to Canarin PM_2.5_ measurements in underground stations (a higher correction factor should be applied for PM_10_). However, no correction was made because it was difficult to determine the precise times when participants were in these environments (e.g., underground railway stations). Nevertheless, participants spent a total of 11.5 h in the subway (which was only 0.78% of the total time), and of this time, only a tiny fraction was spent in the station itself. The time spent in the subway was relatively short because most of the participants lived in Versailles, and the subway was mostly used by the few participants who worked in the city of Paris.

### 2.3. Data Analysis

The data analysis was performed in three main steps: pre-processing, environment assignment, and validation. 

#### 2.3.1. Pre-Processing Phase

Portable sensor measurements are often noisy and may contain outliers, which inevitably biases the analysis of raw data. The pre-processing step aims to detect and eliminate as many artifacts as possible. We implemented an algorithm to detect and remove such artifacts by adapting the peak detection approach [38]. We set up an algorithm that detects when a sudden peak increase occurs in the values of the time series such that its difference with the preceding and following values within a time window is exceptionally high compared to their average. Precisely, s_i_ at the time i is a peak if s_i_ > f* mean ({s_j_ | i-k ≤ j ≤ i + k, j ≠ i}) where f is a given factor, and k determines the window size around s_i_. We empirically set k and f to 2, corresponding to a time window of five minutes and peaks more than twice the mean values. In addition, pre-selected threshold values were implemented in the algorithm so that the out-of-range values were removed (see Appendix A). Some other rule-based processing was performed. For the PM data, there were some inconsistencies, so a condition was added to respect the fact that PM_1_ ≤ PM_2.5_ ≤ PM_10_. The data that did not meet this condition were discarded. The data were also verified by generating graphs by week or day for visual verification. This further control helped detect whether the first-minute data given by the Cairsens were out of line. This was probably due to the device’s heating, so the first three minutes of data were removed for the Cairsens sensors. During this phase, about 10% of the data (with a maximum of 11% for PM_10_) have been deleted.

The AE51 data contained some negative values, unlike the other two sensors. We chose not to rule out the low negative values in order to keep track of the measurement variability. The detection limit (LOD) of the sensor was evaluated at about 1500 ng·m^−3^, corresponding to three times the standard deviation of the signal measured during a period corresponding to background levels. We set the negative threshold of the instrument at −1500 ng·m^−3^ in relation to the LOD.

GPS data is also imprecise due to noise in GPS signals. We implemented a basic GPS data denoising by calculating motion speed between two consecutive points. If it exceeds a certain threshold, the second point is removed (we used a threshold of 130 km as the campaign was conducted in urban and suburban areas). Once denoised, we computed the mean speed per minute and used the generated speed time series in the next step. Subsequently, the GPS coordinate subset that matched the retained sensor measurements were added, drastically reducing the data volume due to the difference between the GPS sampling rate (almost 1 Hz) and the per-minute sampling of the other sensors.

The pre-processing phase ended with the data fusion in a single table containing all the time series. Notice that a row is maintained in this table whenever it has at least one sensor measurement. The labels are added to the dataset for training the machine learning model. Since home and office generate much more samples in the table than outdoor and transport activities, this leads to an imbalanced class problem. The imbalanced dataset exhibits a significant problem for the classifiers to be biased towards the majority class. Therefore, techniques of class balancing should be implemented. We used a combination of the under-sampling of the majority classes and the over-sampling of the minority classes based on a data augmentation algorithm. Precisely, we applied the synthetic minority oversampling technique (SMOTE), which under-samples the majority class and over-samples the minority one by randomly generating new samples close to the border of the minority class data (Appendix A) [39]. Then, we apply the time series generative adversarial (TimeGAN) network to generate a more diverse, realistic time series while considering the temporal characteristics of the data in the minority class [40]. In fact, the generative adversarial network (GAN) has shown promising performance among various types of data, including time series [41].

#### 2.3.2. Environment Assignment

PE strongly depends on the environment. For this reason, there is great interest in making exposure analysis context aware. However, context annotation is the most complicated information to collect in a real-life application setting since only a few participants thoroughly reported their activities during the campaign. Therefore, there is a great interest in automatically detecting the context without burdening the participants.

Furthermore, the plots of data collected during the preliminary tests exhibit patterns specific to some environments. Based on this observation, one may consider the time series of sensor data as predictors of the environment. Therefore, we designed a machine learning algorithm and trained a model using a manually annotated dataset with the respective environment. Then, we applied this model to assign the environment to the sensor measurements of the other participants. This leads to solving a classification problem on time series. The overall process is schematized in Figure 1.

In our machine learning algorithm, the model takes the available measurements (i.e., “temperature”, “humidity”, “PM_1_”, ” PM_2.5_”, “PM_10_”, “NO_2_”, “BC”, and “speed”) as inputs and outputs the environment (context) of the participant. 

To build the model, we adopted multi-view learning, which consists of two-stage: a first-level learner is trained on each view (here, each time series) separately, then a meta-learner is trained on the concatenation of the prediction output (both the prediction class and its probability) by the first-level learners [42]. A concrete example is given in Table 2 below. Thus, the meta-learner predicts the environment by combining the results from previous learners, which enhances the global accuracy of the classification. We trained our multi-view learning model and tested it using a part of the Polluscope data collected in a previous campaign on the RECORD cohort [43]. These data were carefully checked using a dedicated interactive tool, manual verification, and corrections. So, it provides a reliable ground truth for the training and evaluation of our results. We obtained 91% accuracy of the model on the testing set.

Please refer to the following paper, El Hafyani et al. (2022), for more details [44].

#### 2.3.3. Post-Processing/Validation

A post-processing phase was added to the machine learning process to enhance the prediction accuracy further. To segment the stops, we partitioned the data points according to a grid and calculated the density per grid cell. The stops were the pixels with the higher density. Other a priori rules were employed to correct the misclassified segments by the model. For instance, the home locations are the densest cells between 2 a.m. and 5 a.m. With the post-processing, the accuracy improves to 93.4%.

Some GPS data of this campaign around Versailles were spot-checked using a mapping tool to validate the ML allocations. For instance, if the ML predicts an office, but the GPS coincides with a park, this would indicate a misclassification problem. Averages were calculated by environment and pollutant type. This also allowed the affiliations given by the machine learning (ML) algorithm to be verified.

### 2.4. Measurement from the Air Quality Network

The data were also compared with observational data from the Airparif fixed monitoring stations (Figure 2) to analyze the representativity of outdoor measurements to represent PE. The Airparif stations are positioned to represent the different types of environments (urban/suburban background, rural and road traffic) and to assess the spatial variability of atmospheric pollutants over the IDF region.

Of the Airparif monitoring sites, the Versailles, Gennevilliers, and Paris XIII stations were considered in this study. Located in the southern part of Paris (13th district—48°49′42″ N, 02°21′34″ E—57 m altitude), the Paris XIII urban background station dominates a large public garden (called Parc de Choisy) at approximately 400 m from Place d’Italie (grouping a shopping center and main boulevards). NO_2_, ozone (O_3_), and BC are measured at this station. Located 8 km north-west of the center of Paris, the Gennevilliers urban background station (48°55′48″ N, 2°17′39″ E—29 m altitude) is implemented as a residential area at about 50–60 m on both sides of two main roads. NO_2_, PM_10_, PM_2.5_, and BC measurements are performed at this site. The Gennevilliers and Paris XIII stations are about 20 km north-east of Versailles. The station of Versailles (48°47′58″ N, 02°07′53″ E—125 m altitude) operated by Airparif is representative of a suburban background situation. This site is also located in a residential area with parking facilities near the train station and about 100–200 m from two major departmental roads. These three stations are relevant to evaluate outdoor PE measurements.

### 2.5. Additional Experiments in Specific Environments

To better understand the campaign’s results, additional experiments were conducted with the same sensors used during the campaign (IPI indices Table 1). These experiments were conducted in the regional train (RER), metro, bus, streetcar, three cars (gasoline and diesel), and indoors.

These results are presented in Appendix A and helped to interpret the results from the 2019 Polluscope campaign.

## 3. Results and Discussion

### 3.1. Results of All Participants in the Autumn 2019 Polluscope Campaign

The results presented here are for all participants combined. They, therefore, represent the results measured by the 63 participants during the five weeks of the campaign. Results are presented with different time resolutions, daily mean (to be compared with the WHO recommendations), hourly means and medians (to be representative of PE during an integrated period time), and finally, minutely (to represent PE to high concentrations) (Table 3).

Figure 3 and Table 3 show the results as daily averages. Only the days with more than 20% data completion were considered for representativeness purposes. These results were compared to the daily averages from fixed stations (Versailles, Gennevilliers, Paris XIII) and to the WHO recommendation.

Figure 3 shows that the 75th percentile of the participant PE was lower than the WHO recommendations for the three regulated pollutants (NO_2_, PM_2.5_, PM_10_). For NO_2_, the ones measured by the participants are close to the concentrations measured by the Versailles station, with a mean of 18 µg·m^−3^. The concentrations measured by fixed stations (Airparif) in Paris and Gennevilliers are higher (around 30 µg·m^−3^) due to more significant sources, including higher housing density and heavier traffic, particularly in and near Paris. There is no PM_2.5_ or PM_10_ measurement stations in Versailles, so the participants’ measurements were compared to those of Gennevilliers (located about 20 km from Versailles). For PM_2.5_ data, the median and mean concentrations measured by the participants are close to those measured by Airparif; the mean concentrations for both are around 9 µg·m^−3^. On the other hand, for PM_10_ values, the values measured in Gennevilliers by Airparif are on average 17 µg·m^−3^, while the concentrations measured by the participants are on average 10 µg·m^−3^ [42]. The BC concentrations measured by the participants are generally lower than those measured by the fixed stations (Table 3).

This first analysis shows that the results of this study are close to the concentrations that could be measured in a suburban environment (Versailles) at that time, consistent with the volunteers’ location during the measurements (Figure 4).

The data from the fixed stations (Airparif) were only measured outdoors [42]. However, it is crucial to consider that the measurements of the participants were made in different environments (indoor, outdoor and transport).

Figure 4 and Appendix A represent the map of the NO_2_ and BC measurements, respectively, from all participants. The city of Versailles and its surroundings were zoomed in because it was the place of residence of the majority of the participants.

The highest concentrations were located by major roads, with concentrations between 60 and 100 µg·m^−3^ by the freeways for NO_2_ and between 4500 and more than 10,000 ng·m^−3^ for BC. The variability in concentrations was more significant for national and secondary roads than for highways. This is because car flows are more constant and more numerous on freeways. For national and regional roads, car flows are generally less dense. In the Versailles Park and La Boulie (circled in black in Figure 4), concentrations ranged between 0 and 20 µg·m^−3^ for NO_2_ and between 0 and 2000 ng·m^−3^ for BC. These locations are relatively far away from major roads, and the impact of road traffic on the measured concentrations is, therefore, low. Moreover, NO_2_ and BC are primary pollutants with short lifetimes, so they only have little time to accumulate in the atmosphere [43,44]. Away from the source, concentrations decrease rapidly. In Versailles, concentrations varied between 0 and 80 µg·m^−3^ for NO_2_ and between 1500 and 3000 ng·m^−3^ for BC, but they mainly remained low. The concentrations in the city center remained higher than in La Boulie or the Park of Versailles. This is explained by the fact that the emissions of NO_2_ and BC come mainly from road traffic and the residential sector [21,22,45]. As the campaign took place in the autumn of 2019, heating was also used more frequently.

These results provided information on what the participants were exposed to during the campaign. The results show that their exposure was relatively low. For NO_2_, the concentrations from the Versailles station and those measured by the participants during the campaign are similar, which suggest that in some conditions, the outdoor concentration monitored by the network are representative of the mean exposure of citizens.

The results were then separated according to the different environments to understand better where the participants were most exposed.

### 3.2. Results for the Participants According to the Environments

Based on the ML methodology described in Section 2.3. we separated the concentrations measured in five different environments: home, office, indoor (e.g., restaurant, train station, store), outdoor, and transport (e.g., car, train, subway).

Additional experiments were performed to better understand the concentrations and variability measured in the different environments. The results of the experiments performed in a kitchen, car, and subway are presented in Appendix A.

The case study presented below shows the results of the ML and the concentrations of one participant measured during their whole week of participation. Then, the results of all participants combined according to the different environments are presented.

#### 3.2.1. Case Study

The time series of pollutant concentrations measured by the sensors worn by participant 71 is presented in Figure 5. The bottom panel represents the environments reported by the participant, while the other panels (PM_2.5_, NO_2_, and BC) display the pollutant concentrations with the environments recalculated by the ML. The figure shows that the reported environment is absent at the beginning (in white) but that the algorithm recovered it from some PM_2.5_ values, temperature, and humidity time series (not shown in this graph). We can also see that the model predictions are more reliable than the reported environment. For example, the participant reported about 10 h of transport on November 1, which is unlikely. This was replaced by a sequence of short transport and indoor environments episodes.

According to the allowances given by the ML, the concentrations measured at home and the office are relatively low. However, pollution peaks are sometimes observed due to specific activities (e.g., cooking, wood heating) (experiment Appendix A and [28]). The participant spent the majority of their time in these two environments. However, the concentrations measured indoors are pretty high, this participant potentially frequented restaurants, and cooking is an emitter of PM, which explains these concentrations (indoor experiment and [45]). Finally, the concentrations of NO_2_ and BC are relatively high as the participant mainly uses his car as a means of transportation. A substantial increase in NO_2_ and BC in combustion cars is generally observed (experiments in Appendix A and [21,46]). BC concentrations are incredibly high during traffic jams, as seen already in a study of near-road exposure to BC in Korea [47].

#### 3.2.2. Results for All Participants

Figure 6 represents the concentrations of the participants according to the environments and the different pollutants.

The office and home are comparable indoor environments, where participants spent more than 90% of their time during the campaign (Figure 6). The concentrations in these two environments are similar in terms of PM and BC. For comparison, Isiugo et al. (2019) measured indoor BC concentrations between September 2015 and August 2017 in 23 homes in the Cincinnati–Kentucky–Indiana region [28]. A mean concentration of about 850 ng·m^−3^ was measured; this value is close to that measured during our survey of 800 ng·m^−3^ [28]. The measured PM concentrations are slightly higher than those measured for offices because some activities (e.g., cooking) can emit PM (indoor experience and [22]). However, NO_2_ concentrations are higher in offices, which can be explained by the fact that they are usually closer to major roads, unlike homes, which are often located in residential areas [47]. Therefore, it is likely that outdoor air influences indoor air [47]. In comparison, in the review by Salonen et al. (2019), the average NO_2_ concentration measured in offices was 25.1 µg·m^−3^. It is very close to the one measured in our campaign at 28 µg·m^−3^ [48].

Contrary to the home and office, the concentrations measured for all the pollutants in the indoor environment are more comparable to the outdoor and transport environments. The concentrations measured for PM are the highest of all environments; this environment contains places like restaurants or train stations with PM emitters (Section 3.1 and [22,36]). Indoor concentrations of BC may come mainly from the restaurant environment.

For transport, the concentrations are close to those measured outside. All types of transport are mixed in this environment, but many participants used their cars. In a study by Mehel on passenger compartment air quality in the Paris region, the results showed that the concentrations measured for NO_2_ in the cabin were lower than those measured outdoors, 117 µg·m^−3^ outdoors and 80 µg·m^−3^ in the cabin [48]. For PM_1_ concentrations, the study showed that the concentrations are equivalent, i.e., 25 µg·m^−3^ indoors and 23 µg·m^−3^ outdoors [49]. The study by Mehel also showed that the changes in concentrations indoors and outside the vehicle occurred almost at the same time [48]. The concentrations are not directly comparable because the Mehel study only involved cars on freeways, the ring road, or tunnels, i.e., particularly polluted areas in the Paris region. However, some common points can be highlighted: for NO_2_, the general variability is the same as in the Mehel study. The outdoor concentrations are higher than those for transport, which may be linked to the transport cabin, which protects us from part of the pollution. Indeed, vehicles can be equipped with cabin filters that allow for less polluted air in the car interior. Moreover, measured PM_10_ concentrations may need to be considered (Section 2.2). For PM_1_, the general variability is the same: concentrations measured during the campaign are very close, 8 µg·m^−3^ for transport and 9 µg·m^−3^ for outdoors. For the BC, the concentrations are also higher outdoors.

The concentrations measured outdoors are more representative of the concentrations impacted by road transport because these sensors directly measure the outdoor emissions. For comparison, the average NO_2_ concentrations measured by the station in Versailles were calculated over the same period as the campaign. The mean NO_2_ from October to December 2019 is 18 µg·m^−3^. The difference with the 44 µg·m^−3^ determined during our campaign may be due to the volunteers traveling outside Versailles and closer to Paris, where NO_2_ concentrations were higher. The Versailles station is a background station, so it is not directly influenced by road traffic. Over the campaign period, the average NO_2_ concentration measured in a background station in Paris was 31 µg·m^−3^. To compare with another city studied in the literature, in a study conducted by Garcia et al. in Lisbon, Portugal, the average NO_2_ concentrations measured ranged from 29 µg·m^−3^ to 47 µg·m^−3^ [50]. The concentrations are comparable to those measured during the campaign (44 µg·m^−3^) and reflect that it is an urban environment for both studies.

## 4. Conclusions

This study assessed PE to NO_2_, PM_1_, PM_2.5_, PM_10_, and BC. These pollutants were measured using portable sensors that 63 participants wore during a campaign that took place in autumn 2019 in the IDF region.

To analyze the participants’ data, a study of the temporal and spatial variabilities was carried out, as well as analyses according to concentration, pollutants, and environments. The results of this campaign showed that concentrations measured by the participants were very dependent on their environments and activities. An ML algorithm was used to perform the participant allocation. These allocations were calculated based on PM concentrations, temperature, and measured humidities by the Canarin. The allocations calculated by the ML were more reliable than those reported by the participants. NO_2_ concentrations in transport were high (mean of 45 µg·m^−3^), even though participants spent little time there (4% on average). In contrast, participants spent most of their time at home (68%), but the concentrations in this environment were generally low (9 µg·m^−3^ for PM_10_ and 18 µg·m^−3^ for NO_2_). However, indoor pollution peaks caused by various activities (e.g., cooking) were observed. This study also showed that using portable sensors is of great interest. However, these measurements must be combined with a powerful data processing system. In contrast to fixed instruments, hand-held sensors allow participants to be followed in all the environments they encounter, thus complementing existing methods. This made possible the measurement of the PE, which is very dependent on each person’s lifestyle.

Other campaigns conducted at different seasons could provide more thorough data concerning PE. In addition, improving the ML to have a more precise environment allocation could improve data collection. Measurements of other pollutants, such as VOCs, could also improve the characterization of indoor air exposition.

## Figures and Tables

**Figure 1 toxics-11-00206-f001:**
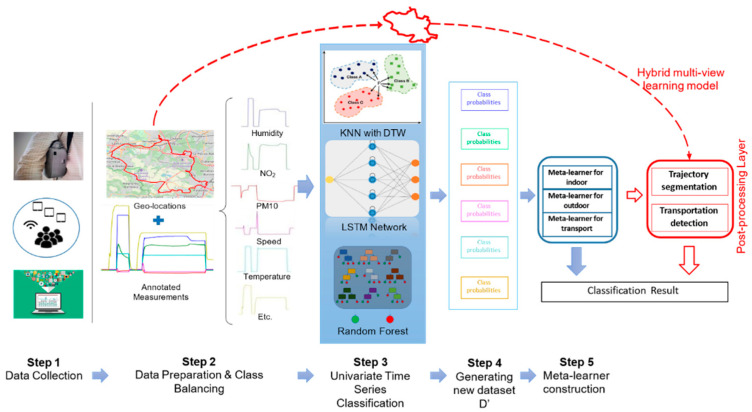
Overview of the environment recognition process.

**Figure 2 toxics-11-00206-f002:**
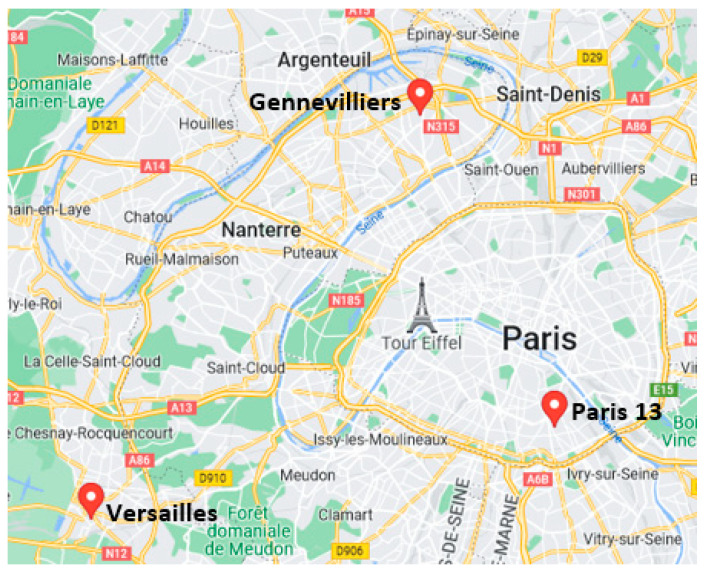
Maps of Airparif stations.

**Figure 3 toxics-11-00206-f003:**
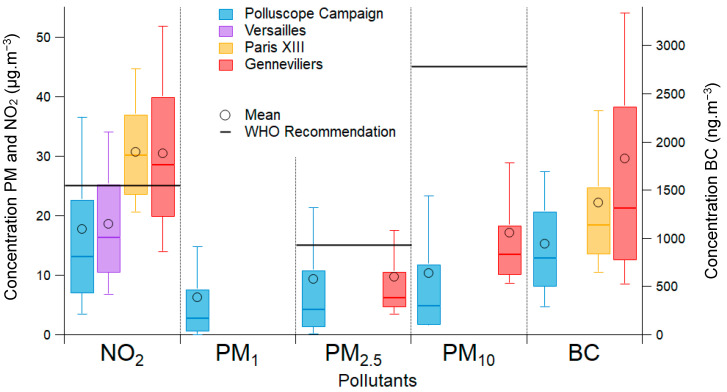
Boxplot of the concentrations measured during the Polluscope campaign and measurements made by the fixed stations of Airparif (Gennevilliers, Paris XIII, Versailles), all data were used with the daily average. The whiskers extend from p10 to p90, the box from p25 to p75, and the median is plotted by the middle line. The black circle represents the mean. The WHO recommendation value is also shown.

**Figure 4 toxics-11-00206-f004:**
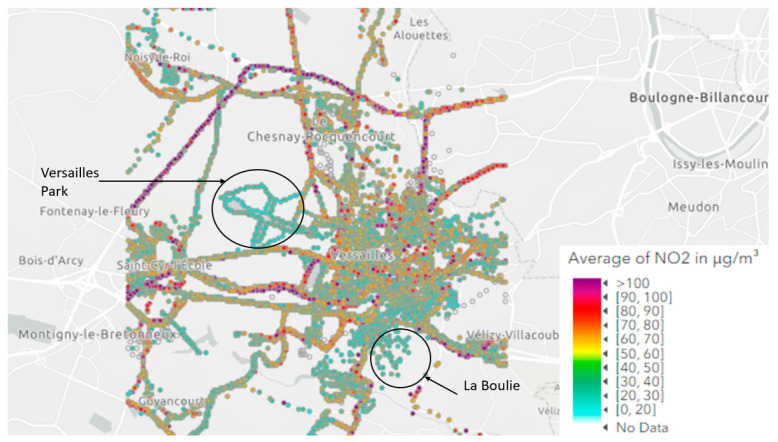
Geo-located NO2 concentration in Versailles.

**Figure 5 toxics-11-00206-f005:**
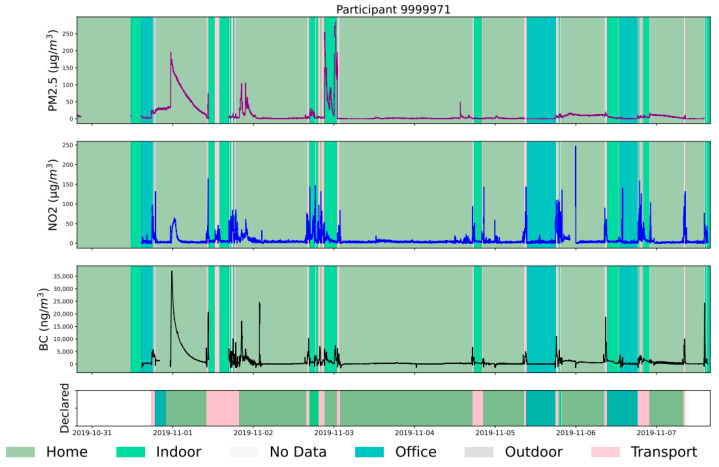
Time plot for participant 71. In black BC (ng·m^−^^3^), in blue NO_2_ in (µg·m^−^^3^) and in purple PM_2.5_ (µg·m^−^^3^).

**Figure 6 toxics-11-00206-f006:**
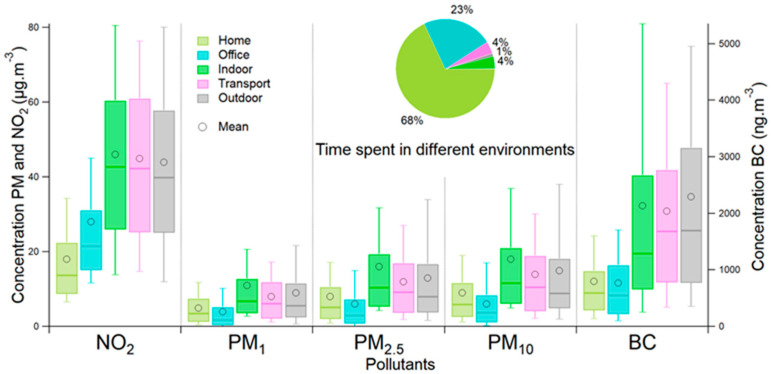
Boxplot of concentrations measured during the Polluscope campaign as a function of different environments. The whiskers extend from p10 to p90, the box from p25 to p75, and the median is represented by the central line. The black circle represents the mean. The pie chart represents the time spent in the different environments.

**Table 1 toxics-11-00206-t001:** Average IPI indices of the sensors.

Sensors	Pollutants	Average IPI Index Campaign 2019 Qualification	Average Pearson Correlation Coefficients
AE51	BC	0.75	0.83
Cairsens	NO_2_	0.69	0.66
Canarin II	PM_1_	0.83	0.93
	PM_2.5_	0.82	0.93
	PM_10_	0.75	0.83

**Table 2 toxics-11-00206-t002:** Input raw data of the meta-learner. This example shows the output of the first-level learners on four views (temperature, humidity, speed, and NO_2_), the predicted environment, and the probability. For instance, the first-level learner predicted “transport” with a probability of 0.6.

First-Level Learners				Associated Prediction Probabilities				True Label
Temperature	Humidity	Speed	NO_2_	Temperature	Humidity	Speed	NO_2_	
Indoor	Indoor	Outdoor	Transport	0.6	0.7	0.5	0.6	Indoor

**Table 3 toxics-11-00206-t003:** Summary of the concentrations for all participants.

		PM_1_ (µg·m^−3^)	PM_2.5_ (µg·m^−3^)	PM_10_ (µg·m^−3^)	NO_2_ (µg·m^−3^)	BC (ng·m^−3^)
Day	Mean	7	11	13	16	846
Hour	Mean (STD)	6 (8)	10 (12)	11 (14)	17 (10)	769 (764)
	Median	3	5	6	15	528
Minute	p95	22	32	35	28	2844

## Data Availability

For data, please contact the corresponding author.

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
