# Peer review of "NO2, BC and PM Exposure of Participants in the Polluscope Autumn 2019 Campaign in the Paris Region"

_toxics, 2023, doi:10.3390/toxics11030206_

Round 1

Reviewer 1 Report

Review: Personal exposure to NO2, BC and PM of participants in the Polluscope autumn 2019 campaign in the Paris region 

This paper describes personal exposure to different pollutants measured via low-cost sensors obtained during the polluscope campaign (2019). The paper describes measurements obtained during this field campaign along with punctual measurements performed in specific environment (indoor/car/subway). The low-cost sensors are well described however the sensor qualification test and the Data analysis sections seem to be too short to be clearly understandable. The result section starts with the presentation of the measurements within the specific environments. The authors claim that those experiments were conducted to improve the machine learning and the interpretation of the results. However, the improvements are not shown and are not evident for the readers. Finally, the personal exposure of the different volunteers is described in three sections (sections 3.2 and 3.3 with all participant measurements and section 3.4 for case study of individual participant). A similar study has been published in the same journal by Languille et al. (2022) with similar author list. This paper is never cited in this manuscript and show personal exposure from Polluscope volunteers. The results highlighted in Languille et al. (2022) show personal exposure of similar pollutants in specific environments (Outdoor/Indoor/Commuting for different seasons) therefore highly similar to what is done in this manuscript probably even more detailed. Due to multiples errors and imprecisions within the manuscript, I suggest major revisions. 

Major revisions: 

The authors often based their work on previous published studies without mentioning the content of those studies. For example, the model to identify the type of environment where the volunteer is performing measurements is not detailed at all. By reading this manuscript, we can’t tell which parameters are used to do so. One would expect to know at least the parameters used and the general methodology to be able to judge the results shown later on. 

The “sensor qualification tests” section is quite light to me. You present IPI as well as numerous tables within the supplementary materials (all different from Languille et al., 2022) but we still don’t know: 

- 1. Where was performed the comparison campaign? Was it done in a real atmosphere on the SIRTA super site or was it done in a controlled environment? 

- 2. What about RH correction of these sensors? Where are the sensors dedicated to measure the RH and T? Is it close to the CANARIN II sensor? How the sensor is modifying the temperature and the humidity? what kind of RH corrections did you use? Is it the same for PM1, PM2.5 and PM10? By experience the concentration can be divided by 2 at high RH, can you state the range of RH correction factors that you encountered? 

- 3. Of course the aerosols observed within the railway stations are denser than within the atmosphere, and therefore could lead to a large underestimation of the mass concentration. You stated that no corrections were applied on the data due to the difficulty to differentiate between different mode of transport. You need, at one point, to estimate the errors that could be associated to the density variability. 

GPS data : You claim that the ML allocation is validated by the analysis of the GPS data set. How accurate is the geolocalisation using your GPS data set ? Is it the only criteria to validate the ML ? Moreover, you are talking about a visual verification (L216) : What does that mean ? How accurate is this visual verification?

Case study : Why is that interesting to this paper to show individual cases ? Of course, there are different people so there will be different way of living and different personal exposure. So, it is expected that from one to another volunteer the pollution levels they are exposed to is changing. The goal of these studies would be more to have a robust data set to statistically analyze the mean or median personal exposure. Instead of showing 2 individual measurements you should probably add to your work (Table 3) some quartiles that would improve greatly the message. 

inor revisions: 

-Please check the numeration of the different section. You have a 3.4.1 and no 3.4.2.

-Not all the references are included in the reference list and not all of them are in the same format. For example, Petit et al (2021) is cited Line 155. 

-L183 : The Aethalometer shows negative values. You stated that a threshold for negative values was set. What is this threshold? With the classic aethalometer, an average is recommended over a specific period of time. How do you fix the negative values? ‘However, weakly negative and weakly positive values related to the instrumental background were retained [37] ‘  What does it mean ? 

- L209 : What denoising the GPS data means ? 

- L213 : “Hourly averages  by environment “ I doubt that the volunteers stay in each environment at least an hour. So how can you get hourly averages? Did you stack all the data and then perform an hourly averages (60 data points)?

- Can you comment on the hourly averages when considering moving volunteers? 

Section 3.1 : I clearly not see the point/interest of these sections in this manuscript. You stated that this is for the machine learning improvement but you never show how. Please remove all these from the main text. 

-Figure 2 label needs to be rephrased... 

-L301 : There are sources of NO2 in the metro. Indeed, work related machines working at night are often using combustion engine. You can’t state that there is no NO2 source especially since all the pollutant sources within the metro (railway or similar environments) are not yet clearly established. 

Figure 4 : BC concentration is then larger than PM1, PM2.5 and PM10 or the specific scale for BC is missing. Given the fact that you seems to justify the extremely large concentration of BC (here it would reach 30ug/m3 ??? ), I believe that the scale is not missing.  So, it means that BC could be found in particle with diameter larger than 10um. The literature is quite explicit about that, BC is mostly found within the PM1. Can you therefore explain your results? 

Table 2 : What is p95 ? 

Table 3: As we don’t clearly know how the ML is working it’s quite hard to get the difference between indoor and Home/office. Apparently, you assumed that it’s not home/office due to larger concentration of PM. How can you be sure that it’s not Home or office when a PM source, like cooking, is on ?  

All these measurements seems really low 

Reviewer 2 Report

The manuscript " Personal exposure to NO2, BC and PM of participants in the  Polluscope autumn 2019 campaign in the Paris region" was done to understand personal exposure in the Paris region. The contribution of this study is of great significance to the current literature where there are not many similar studies on the subject area in France. The structure is well presented for good readability. There are some minor comments that I have in the methods section that needs to be addressed.

Introduction:

The introduction is well written. The authors justify why such a study is needed in France and how it contributes significantly to the current literature.

Methods

Line 182-183 : why was a threshold set and what is the impact of the negative values on this study. The authors should consider briefly explaining the effects of negative values in such a study

Line 185-191: Authors should consider moving to the results section.

Line 197 -207: Authors should consider moving to the discussion/conclusion

Line 211-212: “The data that did not meet this condition were discarded.” Can the authors provide how many data were discarded for transparency?

Line 304: please add “be” after the word “may”

Reviewer 3 Report

Authors of this study monitored personal exposure to NO2, black carbon, and PM1, PM2.5, and PM10 airborne particles of 63 participants living and commuting in central areas in Paris, France. I agree that this topic is worthwhile, but I am critical about this work due to the following reasons:

- The authors fueled the motivation by mixing up environmental measures set by WHO and personal exposure monitoring, which are two separate things never meant to be compared side-by-side or replace each other. The entire intro should be reconsidered. It is not clear what the main goal of this specific campaign has been and why these specific set of air pollutants were opted. There is too much simplification when they tried to motivate the readers in Lines 60 to 87 with false statements have been presented there.

- It is not clear what the principle of operations of the utilized sensors have been. They should briefly explain. Measurement of PM1 to report mass-based concentrations should not be reliable. Such monitors should be very sensitive and easily record false values. It is not clear what the exact procedure/protocol of their personal sampling has been. Where were the sensors located and how comparable the 63 participants, in terms of their living/commute environments were they?

- It is not clear how the results presented in Figures 3 and 4 were averaged over multiple days or multiple participants. Again, since it is not clear how comparable the participants lifestyle has been, it is not clear how these results may be useful. I wonder why there is not error bars displayed on these plots. Figure 6 seems more interesting and novel but still it suffers from the same issues. What is the significant of reporting minute-based concentrations in Table 2?

- Some English edits are required, and some patches of the manuscript are hard-to-understand.  NO2 concentration is commonly presented and standardized in ppb, rather than ug/m3.

- The take-away lessons and findings from this research and the level of generalizability of them are under-question. Contribution of the authors toward the existing gap of knowledge or state-of-knowledge does not seem satisfactory.

Round 2

Reviewer 1 Report

Review: Personal exposure to NO2, BC and PM of participants in the Polluscope autumn 2019 campaign in the Paris region 

The authors have significantly improved the quality of the paper since the first review. Some points still need some modifications before publication. Also please read the paper to correct all the English mistakes. 

L93 and 94: here you are talking about activities. How do you know for sure that the volunteers are either cooking or smoking a cigarette? How many volunteers were selected? for how long? These percentages mean nothing without background information.  

Also, I’m not completely sure I understood what you meant. You wrote that 44% of the PE (for particulate pollutant only) is due to smoking and cooking. How do you estimate the percentage of the PE per activities? I believe that your volunteers are not smoking or cooking 44% of their time so it probably means that you used the PM values. Then, does it mean that 44% of the largest PM values were due to activities such as cooking and tobacco? 

L 203 – 212 : The PM measurements are not corrected from RH. The authors argue that RH was not an issue as it ranges up to 54%. Can you please try to evaluate the errors that can be made due to this RH on PM concentration? Please state somewhere this discussion about RH within the manuscript. Avoiding talking about it make it even more suspicious. 

Please correct the sentences L229-230. 

 L 211: I’m surprised by the total time spent within the subway. I was expecting Parisians to spend at least and on average an hour per day. It raises the question of how did you choose the volunteers ? Are they living and working in the city ? 

L 255 – 260 : what is the data augmentation algorithm you used ? From what I understood, this kind of algorithm is usually creating data in the imbalanced class. Here you stated that you removed 10%of the data. So you removed data from the home and office classes ? 

 Please let the reader know how you did exactly ? 

Minor comments

L39 : Monitoring concentrations not “ monitoring of the “

L49 : Many studies have  

L85 : BC sources include= ??? 

L159 : Please change ‘Capacibilities ‘

L242 – 243 : So the negative threshold was set at ??? 

L499 – 501 : please rephrase  “The sensors are in direct contact with the emissions  “

Reviewer 3 Report

Authors have addressed my comments acceptably. Please note that quality of Table 2 is much lower than the other tables and the text, which I wonder why. Cincinnati is not a US state. It is a city. I wonder what they meant by Cincinnati-Kentucky-Indiana states.
